# Population Structure and Reproductive Biology of the Endangered Crab *Deiratonotus japonicus* (Brachyura, Camptandriidae) Surveyed for Nine Years in the Kita River, Japan

**Il-Kweun Oh *** and **Seung-Woo Lee**

Graduate School of Environmental Engineering, University of Kitakyushu, 1-1 Hibikino, Wakamatsu, Kitakyushu, Fukuoka 808-0135, Japan; leesw@kitakyu-u.ac.jp
* Correspondence: kweuni51@gmail.com; Tel.: +81-93-695-3293

**Abstract:** *Deiratonotus japonicus* (*D. japonicus*) is known as a near-threatened species, because of the changing conditions of its habitat. This species resides in isolated locations and in upstream, brackish waters from Kanagawa Prefecture to Okinawa Prefecture in Japan. In this study, we investigated the population structure and reproductive biology of *D. japonicus* in the Kita River, Japan. The distribution, sex ratio, breeding season, and fecundity were assessed at bimonthly intervals during spring low-tide periods from May 2001 to November 2008 and from November 2014 to January 2016 for approximately nine years. A total of 3525 crabs were collected during the sampling period: 1806 (51.2%) males, 1240 (35.2%) non-ovigerous females, and 479 (13.6%) ovigerous females. The overall sex ratio (1:0.95) did not differ significantly from the expected 1:1 ratio. The mean maximum density was 26.1 and 36.5 indiv./m$^2$ for the first and second sampling periods, respectively, in the sampling station 5.2 km from the Kita River mouth, and all individuals were typically found approximately 4.4–6.8 km (13.2 ± 7.8 indiv./m$^2$) from the Kita River mouth. Carapace width (CW) ranged from 2.6 to 13.5 mm in males and from 2.8 to 13.4 mm in females and was significantly different between the two sexes ($p < 0.05$). Ovigerous females were found almost throughout the entire sampling period, with breeding peaks between July and September. The smallest ovigerous female had a CW of 3.9 mm. The seasonal frequency distribution suggested the continuous recruitment of young juveniles (CW < 3.9 mm) throughout the year, with peaks from September to November. The mean fecundity was 1008.3 ± 183.1 (8.3 ± 1.6 mm) eggs. Egg number in relation to CW was calculated by the formula egg number (EN) = 110.36 × CW + 90.96 ($R^2 = 0.948$, $n = 41$, $p < 0.0001$). Regression analysis showed that fecundity was closely associated with female CW. Our results indicate that the performance of reproductive biology depends not only on continuous breeding but also on recruitment throughout the year in our study area.

**Keywords:** *Deiratonotus japonicus*; camptandriidae; population structure; reproductive biology; sex ratio; fecundity

## 1. Introduction

The family Camptandriidae currently includes 40 species and 22 genera from South Africa, West Africa, New Caledonia, and Northeast Asia. Most species of Camptandriidae live in marine, intertidal estuarine, and mangrove habitats [1–3]. Camptandriidae were previously classified as a subfamily of Ocypodidae, but now they are classified as a distinct family [4–6]. The Camptandriidae genus *Deiratonotus* Manning & Holthuis, 1981, contains three species: *Deiratonotus cristatus* (De Man, 1895), *Deiratonotus japonicus* (Sakai, 1934), and *Deiratonotus kaoriae* (Miura, Kawane & Wada, 2007) [6,7].

*D. japonicus* (Sakai, 1934) and *Deiratonotus tondensis* (Sakai, 1983) are classified as the same species, according to Kawane et al. [5].

　　*D. japonicus* (Sakai, 1934) (Decapoda: Camptandriidae) is an endemic crab species and has been classified as a near-threatened species in the Red Data Book List by the Ministry of the Environment in Japan. It is usually found in isolated locations and in upstream, brackish waters. Records of *D. japonicus* are limited to several localities from Kanagawa Prefecture to Okinawa Prefecture in western Japan [5,8–10]. Several aspects of the biology and behavior of *D. japonicus* (Sakai, 1934) have been studied: Yamanishi et al. [8,11] and Hiu et al. [12–14] investigated its habitat characteristics and behavior associated with water levels, salinity, and sediment; Terada described its early zoeal stages [4]; Oh et al. observed its larval development under laboratory conditions [10]; and Kawane et al. [5] and Miura et al. [7] investigated its morphological and genetic characteristics.

　　Despite the relative abundance of studies on the general biology of Ocypodidae crabs, biological information on these species is limited. Our study represents the first attempt to clearly identify the population and reproductive biology of *D. japonicus* in the Kita River; no previous studies have been published on this species. We investigated the distribution of size frequency, sex ratio, breeding period, juvenile recruitment, and fecundity of *D. japonicus* in the Kita River, Japan. This study provides information on the biological characteristics, distribution, and reproductive biology of *D. japonicus*, which will also be useful in developing future conservation strategies for its specific habitat and for environmental evaluations in the Kita River, Japan.

## 2. Materials and Methods

　　The study area is located in the Kita River estuary, Miyazaki Prefecture, Japan (32°35′26″ N, 131°42′50″ E, Figure 1). Field sampling for *D. japonicus* was performed at bimonthly intervals from May 2001 to November 2008 and from November 2014 to January 2016. Specimens were collected at 38 stations (including both left and right points at intervals of 400 m from the Kita river mouth to 7 km upstream) during the spring low tide, using a quadrat trap with a net (0.5 m × 0.5 m × 0.3 m). Generally, passive sampling methods using a trap feeding bait have been broadly used. However, in this study no bait was used for sampling to help reduce the sampling time and understand the natural habitats of *D. japonicus* [8,10–14]. The mean density of the crabs captured at each station was obtained by averaging the number of individuals per unit area where 2 to 5 collections were performed. Normalized traps of the same dimensions were used throughout all samplings to ensure the accuracy of sampling.

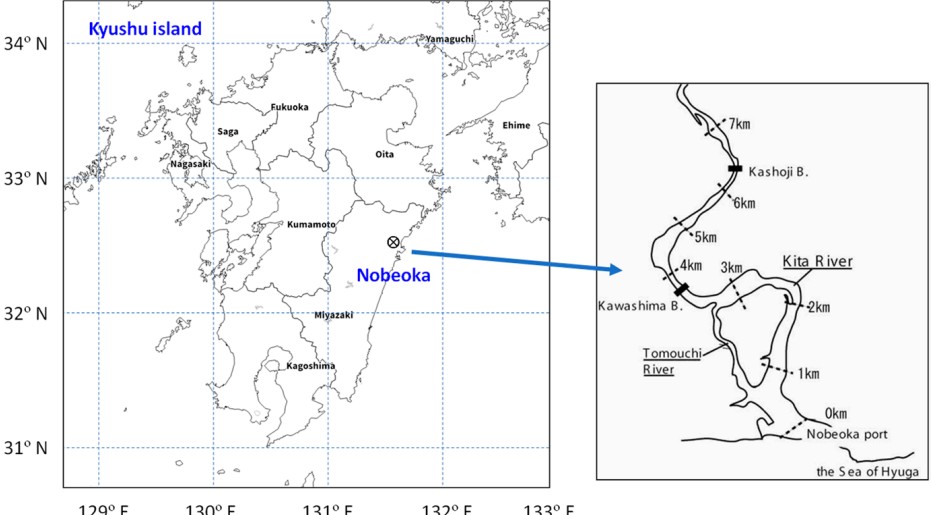

**Figure 1.** Map showing the sampling stations of *Deiratonotus japonicus* from the Kita River estuary.

The habitat characteristics, body size defined as carapace width (CW, in mm), and sex, including maturity and ovigerous condition (mainly related to the temperature or salinity of the habitat environment), were recorded for female specimens. CW was measured by using Vernier calipers to approximately 0.05 mm. Sex was verified by the gonopore position and abdomen characteristics. Ovigerous females were identified by the presence of eggs on their abdomen. The population size structure was analyzed as a function of the size-frequency distribution of all individuals collected during the study period. All crabs captured for this study were released after checking their carapace width, sex, and the presence of eggs, except for some ovigerous females used for fecundity analysis. Specimens were grouped in 1.0 mm-size classes, from 1.0 to 14.0 mm CW. All female specimens smaller than the smallest ovigerous female captured (CW = 3.9 mm) were categorized as juveniles [9,15–17]. Seasonal juvenile recruitment was evaluated by the juvenile proportion. *D. japonicus* reproduction occurs generally throughout the year, with the maximum number of ovigerous females occurring in the summer [10–13]. As the embryos mature, the egg color changes from bright orange to dark brown. To estimate fecundity, *D. japonicus* ovigerous females were collected at the 5.75 km mark of the Kita River in July and September 2015. Ovigerous females were transported in aerated 3 L containers to a laboratory in Kitakyushu University, Fukuoka, Japan. In the laboratory, the CW of ovigerous females was measured, and the number of eggs at stage I (bright orange) was counted with a method proposed by Litulo [15] and Nakayama et al. [18]. Fecundity analysis was conducted with 41 ovigerous females [19–22]. The breeding period was determined by observing ovigerous female frequency data throughout all the sampling periods. Descriptive statistics were calculated for males, females, and ovigerous females for distribution comparison. The chi-squared ($\chi^2$) test was used to assess the sex ratio and to compare the male and female percentages. One-way analysis of variance (ANOVA) was performed to test significant differences among groups. Tukey's and Fisher's tests were also performed. The Kolmogorov–Smirnov (KS) test was used to test the null hypothesis that male and female samples had similar distributions. Differences between treatments were considered significant at $\alpha = 0.05$ for all statistical tests. For fecundity analysis, the data were analyzed using the power function ($y = a\chi + b$) of the egg number (EN) vs. the CW [16,22,23]. All statistical analyses were conducted using the XLSTAT statistical package (version 2014.5.03; Addinsoft, NY, USA).

## 3. Results

### 3.1. Population Structure

#### 3.1.1. Sex Ratio

The overall sex ratio was 1:0.95 (male/female) and did not differ significantly from the expected 1:1 ratio by the chi-squared test ($p > 0.05$, Table 1, Figure 2). The highest abundance of males was found from September to November and decreased from January to July. The seasonal variation of the sex ratio in the sampling periods was analyzed using ANOVA. These results indicated that the sex ratio was significantly different between January and September, as well as in November, with a confidence interval of 95% (Tukey's test, $p < 0.05$) during all the sampling periods. However, the sex ratio did not differ significantly from November 2014 to January 2016 (ANOVA, $p > 0.05$).

**Table 1.** Abundance and sex ratio of *D. japonicus* observed during the sampling periods. Sampling period: May 2001–November 2008 and November 2014–January 2016 (months highlighted in gray).

| Month | Male [1] | | Female | | | | Total | | Sex Ratio |
| | | | Non-Ovigerous | | Ovigerous | | | | |
| | N | % | N | % | N | % | N | % | |
| May | 114 | 50.0 | 114 | 50.0 | - [2] | - | 228 | 6.5 | 1:1.00 |
| July | 32 | 34.8 | 60 | 65.2 | - | - | 92 | 2.6 | 1:1.88 |
| September | 86 | 61.9 | 53 | 38.1 | - | - | 139 | 3.9 | 1:0.62 |

**Table 1.** *Cont.*

| Month | Male [1] N | Male [1] % | Female Non-Ovigerous N | Female Non-Ovigerous % | Female Ovigerous N | Female Ovigerous % | Total N | Total % | Sex Ratio |
|---|---|---|---|---|---|---|---|---|---|
| November | 58 | 55.2 | 47 | 44.8 | - | - | 105 | 3.0 | 1:0.81 |
| January | 40 | 39.6 | 59 | 58.4 | 2 | 2.0 | 101 | 2.9 | 1:1.53 |
| March | 67 | 43.8 | 67 | 43.8 | 19 | 12.4 | 153 | 4.3 | 1:1.28 |
| May | 52 | 41.6 | 33 | 26.4 | 40 | 32.0 | 125 | 3.6 | 1:1.40 |
| July | 56 | 46.3 | 15 | 12.4 | 50 | 41.3 | 121 | 3.4 | 1:1.16 |
| September | 31 | 52.5 | 5 | 8.5 | 23 | 39.0 | 59 | 1.7 | 1:0.90 |
| November | 14 | 51.9 | 13 | 48.2 | 0 | 0.0 | 27 | 0.8 | 1:0.93 |
| January | 27 | 45.0 | 32 | 53.3 | 1 | 1.7 | 60 | 1.7 | 1:1.22 |
| March | 44 | 43.6 | 52 | 51.5 | 5 | 5.0 | 101 | 2.9 | 1:1.30 |
| May | 61 | 46.9 | 45 | 34.6 | 24 | 18.5 | 130 | 3.7 | 1:1.13 |
| July | 31 | 38.8 | 17 | 21.2 | 32 | 40.0 | 80 | 2.3 | 1:1.58 |
| September | 18 | 50.0 | 10 | 27.8 | 8 | 22.2 | 36 | 1.0 | 1:1.00 |
| November | 52 | 59.1 | 30 | 34.1 | 6 | 6.8 | 88 | 2.5 | 1:0.69 |
| January | 54 | 58.7 | 37 | 40.2 | 1 | 1.1 | 92 | 2.6 | 1:0.70 |
| March | 36 | 46.8 | 36 | 46.8 | 5 | 6.5 | 77 | 2.2 | 1:1.14 |
| May | 30 | 39.0 | 31 | 40.3 | 16 | 20.8 | 77 | 2.2 | 1:1.57 |
| July | 35 | 46.7 | 14 | 18.6 | 26 | 34.7 | 75 | 2.1 | 1:1.43 |
| September | 9 | 60.0 | 2 | 13.3 | 4 | 26.7 | 15 | 0.4 | 1:0.67 |
| November | 10 | 66.7 | 5 | 33.3 | 0 | 0.0 | 15 | 0.4 | 1:0.50 |
| January | 4 | 36.4 | 6 | 54.6 | 1 | 9.1 | 11 | 0.3 | 1:1.75 |
| March | 8 | 53.3 | 4 | 26.7 | 3 | 20.0 | 15 | 0.4 | 1:0.88 |
| May | 12 | 52.2 | 4 | 17.4 | 7 | 30.4 | 23 | 0.6 | 1:0.92 |
| July | 7 | 41.2 | 2 | 11.8 | 8 | 47.1 | 17 | 0.5 | 1:1.43 |
| September | 18 | 54.6 | 5 | 15.2 | 10 | 30.3 | 33 | 0.9 | 1:0.83 |
| November | 23 | 62.2 | 14 | 37.8 | 0 | 0.0 | 37 | 1.0 | 1:0.61 |
| January | 32 | 42.1 | 40 | 52.6 | 4 | 5.3 | 76 | 2.2 | 1:1.38 |
| March | 17 | 50.0 | 13 | 38.2 | 4 | 11.8 | 34 | 0.9 | 1:1.00 |
| May | 15 | 42.9 | 8 | 22.9 | 12 | 34.2 | 35 | 1.0 | 1:1.33 |
| July | 12 | 44.4 | 3 | 11.1 | 12 | 44.4 | 27 | 0.8 | 1:1.25 |
| September | 13 | 61.9 | 2 | 9.5 | 6 | 28.6 | 21 | 0.6 | 1:0.62 |
| November | 39 | 56.5 | 30 | 43.5 | 0 | 0.0 | 69 | 2.0 | 1:0.77 |
| January | 35 | 50.0 | 35 | 50.0 | 0 | 0.0 | 70 | 2.0 | 1:1.00 |
| March | 17 | 56.7 | 11 | 36.7 | 2 | 6.6 | 30 | 0.9 | 1:0.77 |
| May | 6 | 40.0 | 5 | 33.3 | 4 | 26.7 | 15 | 0.4 | 1:1.50 |
| July | 17 | 48.6 | 6 | 17.1 | 12 | 34.3 | 35 | 1.0 | 1:1.06 |
| September | 8 | 88.9 | 0 | 0.0 | 1 | 11.1 | 9 | 0.3 | 1:0.13 |
| November | 35 | 77.8 | 9 | 20.0 | 1 | 2.2 | 45 | 1.3 | 1:0.29 |
| January | 64 | 66.7 | 31 | 32.3 | 1 | 1.0 | 96 | 2.7 | 1:0.50 |
| March | 30 | 51.7 | 27 | 46.6 | 1 | 1.7 | 58 | 1.7 | 1:0.93 |
| May | 16 | 66.7 | 5 | 20.8 | 3 | 12.5 | 24 | 0.7 | 1:0.50 |
| July | 18 | 75.0 | 5 | 20.8 | 1 | 4.2 | 24 | 0.7 | 1:0.33 |
| November | 87 | 82.1 | 15 | 14.2 | 4 | 3.7 | 106 | 3.0 | 1:0.22 |
| November | 50 | 54.9 | 35 | 38.5 | 6 | 6.6 | 91 | 2.6 | 1:0.82 |
| January | 43 | 58.9 | 28 | 38.4 | 2 | 2.7 | 73 | 2.1 | 1:0.70 |
| March | 33 | 53.2 | 22 | 35.5 | 7 | 11.3 | 62 | 1.8 | 1:0.88 |
| May | 51 | 51.5 | 23 | 23.2 | 25 | 25.3 | 99 | 2.8 | 1:0.94 |
| July | 38 | 43.2 | 7 | 7.9 | 43 | 48.9 | 88 | 2.5 | 1:1.32 |
| September | 38 | 53.5 | 6 | 8.5 | 27 | 38.0 | 71 | 2.0 | 1:0.87 |
| November | 38 | 47.5 | 35 | 43.7 | 7 | 8.8 | 80 | 2.3 | 1:1.11 |
| January | 25 | 45.5 | 27 | 49.1 | 3 | 5.4 | 55 | 1.6 | 1:1.20 |
| Total | 1806 | 51.2 | 1240 | 35.2 | 479 | 13.6 | 3525 | 100.0 | 1:0.95 |

[1] Classified as male, less than 2.8 mm. [2] Undistinguished between ovigerous and non-ovigerous females.

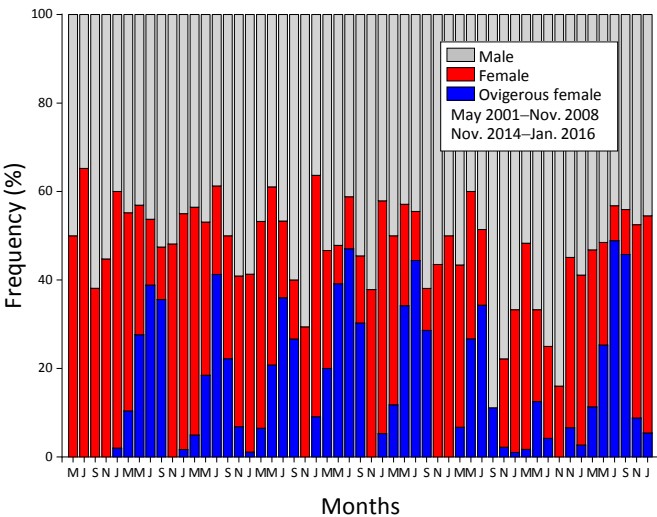

**Figure 2.** Proportions of all individuals of *D. japonicus* during the sampling periods.

### 3.1.2. Distribution and Abundance

We collected *D. japonicus* from the same 38 locations in the Kita River from May 2001 to November 2008 (for 91 months) and from November 2014 to January 2016 (for 15 months) for approximately nine years. Most individuals were collected from the upstream, brackish area. *D. japonicus* individuals were generally collected from, and appeared to be closely associated with, coarse sand and cobble substrate habitats. In these substrate habitats, *D. japonicus* crabs were often found under rocks and rubble or nestled in crevices and corners, as shown in Figure 3. They were generally found approximately 4.4–6.8 km (92.9%) from the Kita River mouth throughout the sampling periods (Figure 4).

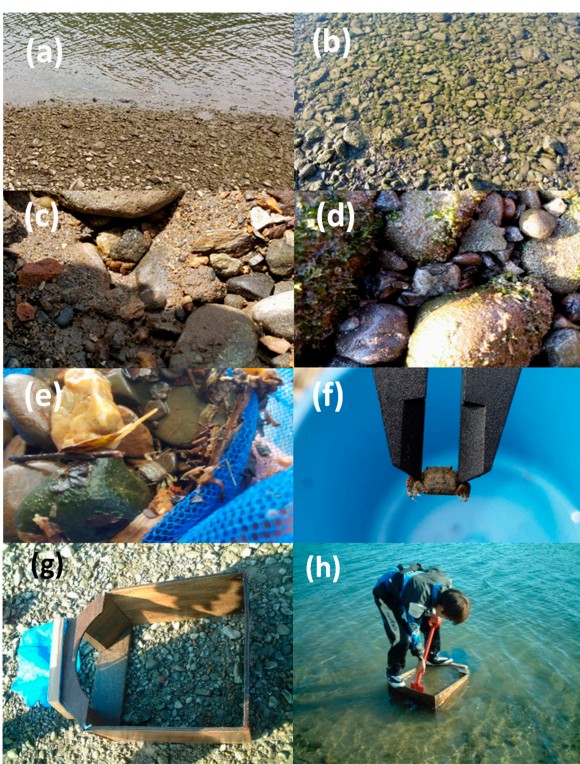

**Figure 3.** Photos of *D. japonicus* sampling stations (5.2–5.6 km from the Kita River mouth): (**a**) 5.2 km habitat point, (**b**) 5.6 km habitat point, (**c**) enlarged image of a, (**d**) enlarged image of b, (**e**) sample collection using a quadrat trap with a net, (**f**) body size measurement using Vernier calipers, (**g**) quadrat trap with a net used in this study, and (**h**) field sampling demonstration.

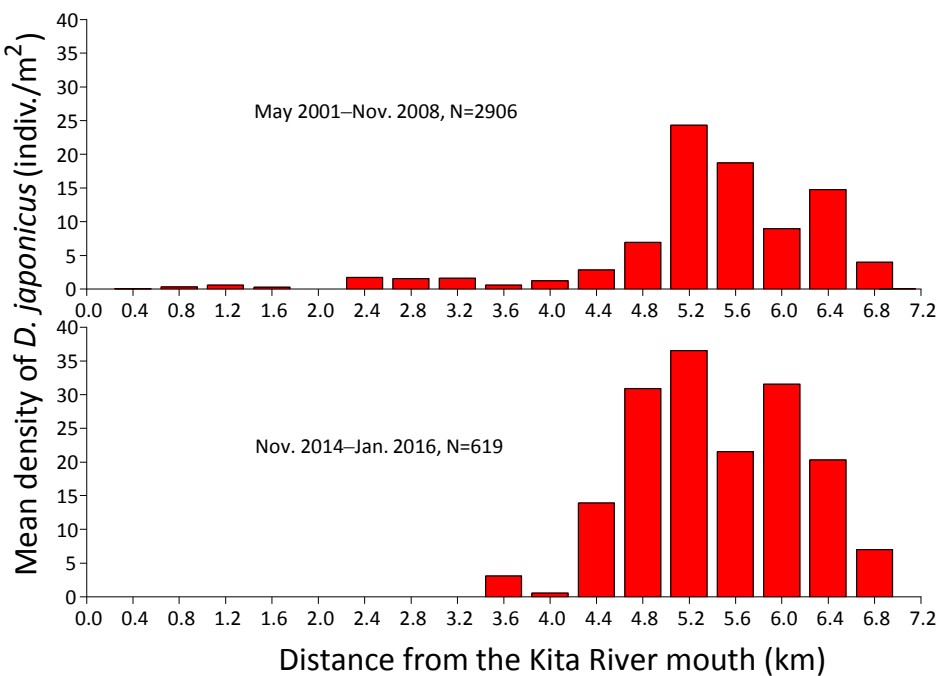

**Figure 4.** Mean density of all individuals by distance from the Kita River mouth.

We analyzed the population abundance and size structure from our sampling data. A total of 3525 crabs were collected during the sampling periods: 1806 males (51.2%), 1240 non-ovigerous females (35.2%), and 479 ovigerous females (13.6%) (Table 1). Differences in the population were observed between the sampling periods. Moreover, there were significant differences between the crab densities and the sampling periods, according to ANOVA ($p < 0.05$, Table 1). The results show that the densities of females (including ovigerous females) were significantly different between November and March and between May and September. In contrast, the densities of males were significantly different only between July and November. The population increased with increasing distance from the river mouth, with a population density ranging from 0.3 to 89.3 indiv./m². The mean maximum density of *D. japonicus* was observed at 5.2 km, showing 26.1 and 36.5 indiv./m² for the first and second sampling periods, respectively (Figure 4). However, the distribution of *D. japonicus* by distance was not significantly different during the sampling periods (KS, $p > 0.05$). An interesting feature is that *D. japonicus* has shown a tendency to slightly move downstream recently when compared with its distribution during the first sampling period (from May 2001 to November 2008), which may be due to some kind of environmental changes such as salinity, temperature, and tide.

The population size structure with respect to CW ranged from 1.6 to 13.5 mm CW, including crabs of no clear sex. Figure 5 shows the size-frequency distribution observed in the male, non-ovigerous female, and ovigerous female populations throughout the sampling periods. CW ranged from 2.6 to 13.5 mm (mean ± standard deviation (SD): 5.4 ± 1.8 mm) for males, from 2.8 to 13.4 mm (mean ± SD: 6.2 ± 1.7 mm) for non-ovigerous females, and from 3.9 to 13.2 mm (mean ± SD: 8.2 ± 1.4 mm) for ovigerous females. The distribution of size frequency was different between the groups (KS, $p < 0.0001$). The maximum size of the males (13.5 mm) was similar to that of the females (13.4 mm).

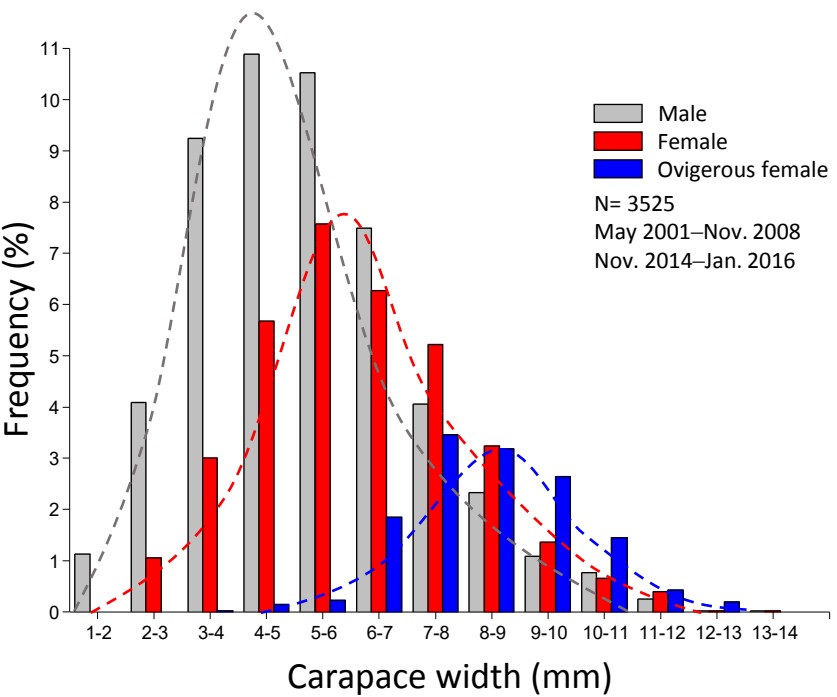

**Figure 5.** Overall size-frequency distribution of *D. japonicus* during the sampling periods.

## 3.2. Reproductive Biology

### 3.2.1. Breeding Period and Recruitment

Ovigerous females were typically collected in the upstream areas. Ovigerous female individuals were found almost throughout the entire sampling period, with the maximum number found between July and September, which appears to be the peak breeding period (Figure 6). A relationship between the frequency of ovigerous females and the mean water temperature was observed; specifically, the ovigerous female population increased with the rising water temperature.

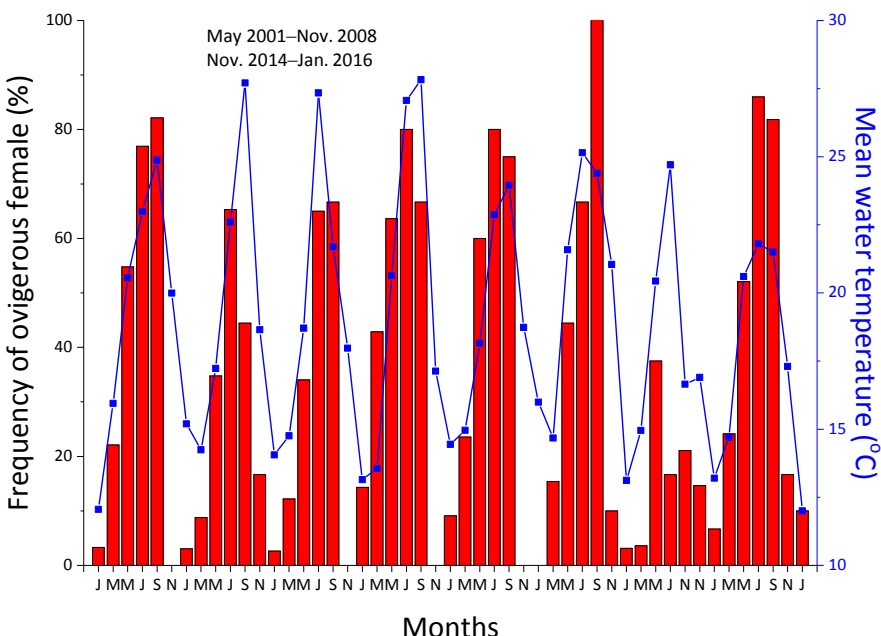

**Figure 6.** The relationship between mean water temperatures (°C) and frequency of ovigerous females during the sampling periods.

These results show that the water temperature significantly influenced the reproductive cycle of *D. japonicus*. Although there appeared to be fewer individuals during winter (mean water temperature, 14.3 ± 1.1 °C), the reproductive cycle was continuous throughout the year, with peaks of occurrence of ovigerous females during the summer (mean water temperature, 24.0 ± 3.0 °C). The mean CW of ovigerous females was 8.2 mm, while the minimum CW was 3.9 mm. Most females had CW between 6.0 and 10.0 mm, with a maximum CW of 13.2 mm (Figure 5). Statistical analysis indicated that the population of ovigerous females was significantly different between the periods of July and September and November and March (ANOVA, *p* < 0.05, Figure 6).

All individuals with CW smaller than the 3.9 mm (the minimum size of ovigerous females) were considered young juveniles. Young juveniles (CW < 3.9 mm) were collected to estimate the possible periods of recruitment. These results suggest a continuous pattern of recruitment throughout the sampling periods. The highest frequency of juveniles was found from September to November, while a clear decrease occurred from January to July (Figure 7).

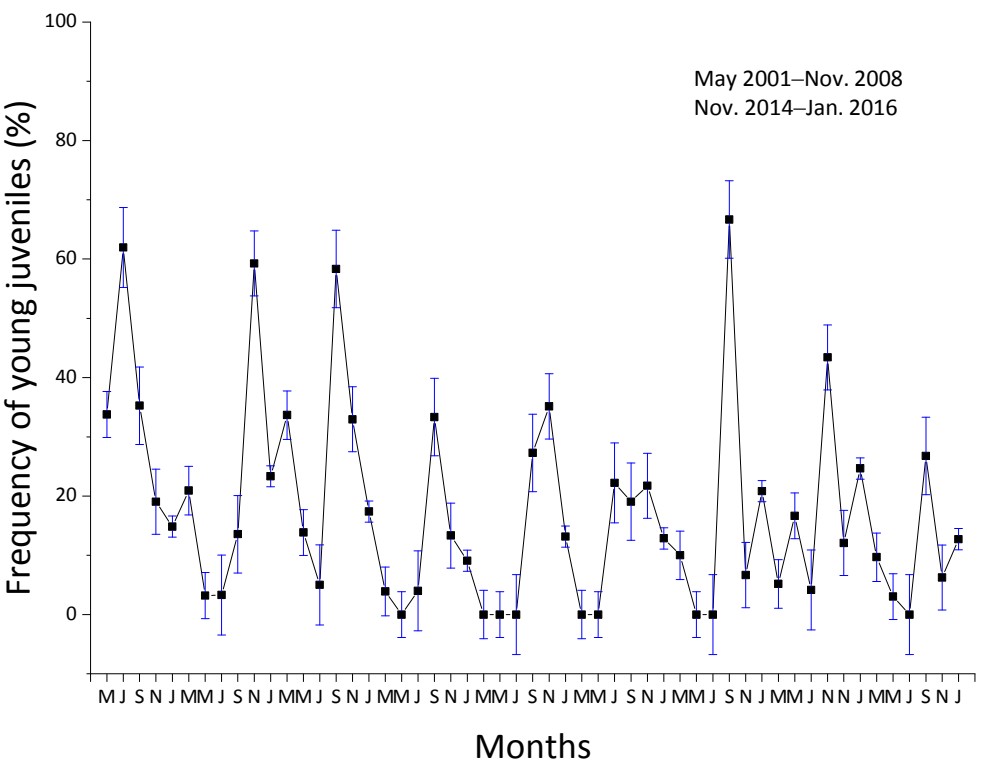

**Figure 7.** Seasonal changes of young juveniles during the sampling periods (bar = ±SD).

The frequency of ovigerous female occurrence in the population appeared to be linked to the absence of juveniles. ANOVA (*p* < 0.05) analyses showed significant differences in juvenile recruits' abundance between September and March, as well as from May to July. Also, there was no difference in the distribution of the ovigerous females and juveniles as the distance from the Kita River mouth increased, according to the Mann–Whitney–Wilcoxon test (*p* > 0.05, Figure 8). This feature is well consistent with the mean maximum density of *D. japonicus*, as shown in Figure 4.

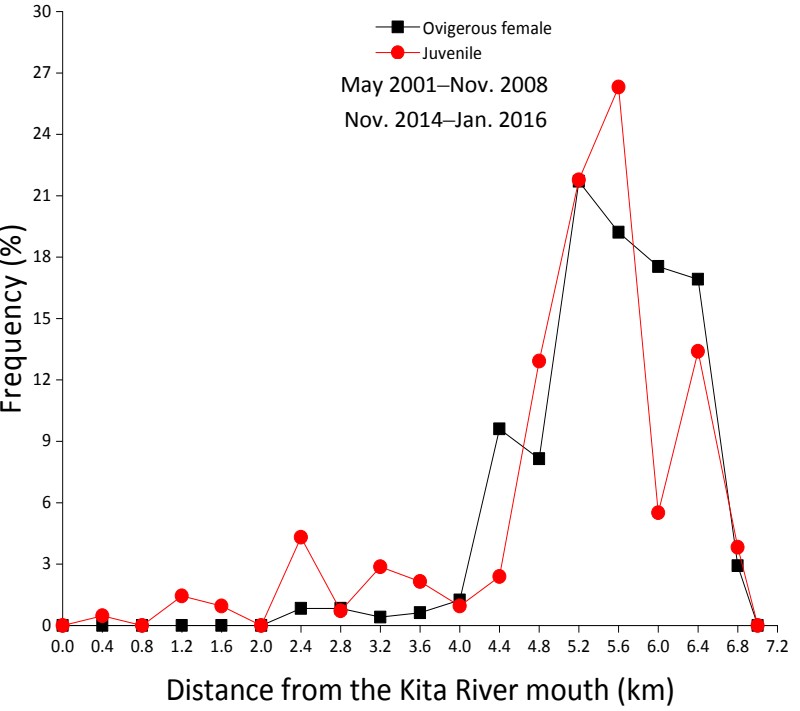

**Figure 8.** Frequency of young juveniles and ovigerous females by distance from the Kita River mouth during the sampling periods.

### 3.2.2. Fecundity

The fecundity of *D. japonicus* ranged from 690 (CW 5.8 mm) to 1359 (CW 11.6 mm) eggs. The mean fecundity was 1008.3 ± 183 (mean ± SD: 8.3 ± 1.6 mm) eggs. The formula to calculate EN in relation to CW could be described as a power function ($y = a\chi + b$). As shown in Figure 9, the number of eggs correlated positively with female size, and the resulting scatter plot shows a linear trend, expressed by the function EN = $110.36 \times$ CW + 90.96 ($R^2 = 0.948$, $n = 41$, $p < 0.0001$). These results suggest that fecundity is closely related to CW in all ovigerous females.

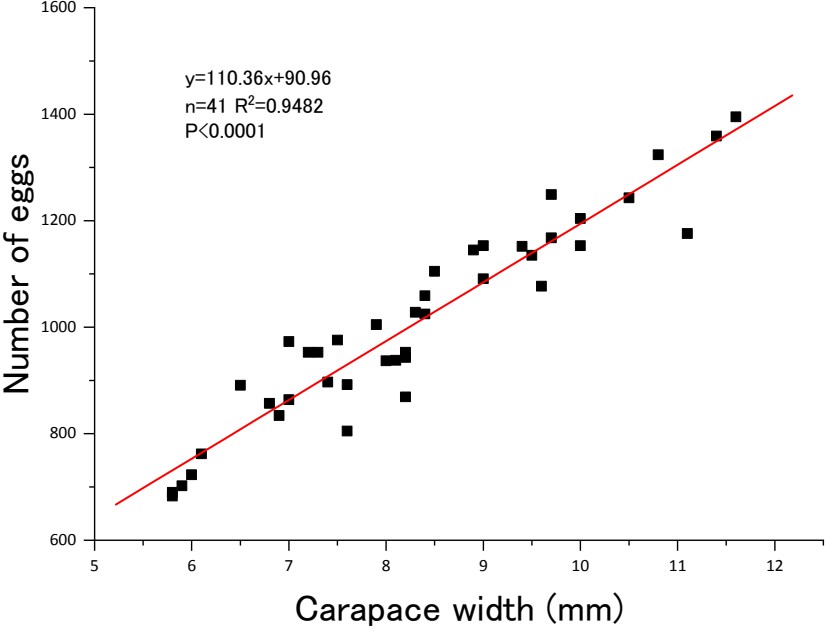

**Figure 9.** Relationship between egg number (EN) and carapace width (CW) of ovigerous females.

## 4. Discussion

*D. japonicus* of the family Camptandriidae typically inhabits intertidal estuarine and mangrove environments [1,8,24,25]. Fukui et al. reported *D. tondensis* in the Tonda River estuary in central Japan [26]. *D. tondensis* individuals were found in upstream, brackish water areas (8.0–25.2 °C, below 10 practical salinity unit: PSU, cobbles). Yamanishi et al. collected all *D. japonicus* individuals within a range of 1.3–6.5 km in the Kita River mouth [11]. Hiu et al. reported the habitat characteristics (4.8–6.4 km, 20–25 °C, 0–10 PSU, sand, and cobbles) in this study area [12,13]. In our study, most of *D. japonicus* individuals were located in limited parts of the upstream, brackish area (4.4–6.8 km mark, 92.9% of the total) in the Kita River estuary during the sampling periods. The habitat of this species is characterized by coarse sand and a cobble substrate at depths of 0.0–1.5 m. The annual mean water temperatures ranged from 12.0 °C to 27.8 °C, and the tidal range was within 1.5 m. The average salinity levels were between 0.0 and 10.0 PSU because of freshwater run-off in rainy seasons.

The overall sex ratio (1:0.95) did not differ significantly from the expected 1:1 ratio. However, significant deviations were observed during the sampling periods. This result corroborates previous studies, which noted that seasonal sex ratios differed significantly from the 1:1 ratio [12]. In crustacean populations, sexual differences in distribution and mortality may be responsible for unbalanced sex ratios [27]. Wenner and Montague suggested that different sex ratios result from different life spans, distributions, migration patterns, growth rates, mortality, and sampling methods [28,29]. Lunar phases and intertidal zonation are known to be determinants of sex ratio variations in brachyuran crabs [30]. The deviation from the 1:1 sex ratio could internally regulate the size of a population by influencing its reproductive potential [30,31]. In our results, the variation in the sex ratios is clearly associated with the spatial–temporal distribution of reproductive activities.

As summarized in Table 1 and Figure 2, the total abundance of crabs during the sampling periods was 3525: 1806 males (51.2%), 1240 non-ovigerous females (35.2%), and 479 ovigerous females (13.6%). The population density of the crabs ranged from 0.3 to 89.3 indiv./m$^2$. The average maximum density was 26.1 indiv./m$^2$ at 5.2 km. Although the population of crabs was somewhat irregular, it increased in size from spring to summer. In the Tonda River, *D. tondensis* (CW > 4.0 mm) was reported to have the highest densities in October and November and the lowest densities in August around the mid- and low-tide areas [26]. Hiu et al. reported that high densities of *D. japonicus* were found in upstream areas from May to September in the Kita River [12,13]. The population size structure according to CW ranged from a CW of 2.6 to a CW of 13.5 mm (mean ± SD: 5.4 ± 1.8 mm) for males and from 2.8 to 13.4 mm (mean ± SD: 6.2 ± 1.7 mm) for females. Litulo and Hartnoll suggested that females reduced somatic growth compared to males because more energy was needed for gonad development and egg production during the reproductive periods [23,32]. In this study, the maximum size of the males (CW 13.5 mm) was similar to that of the females (CW 13.4 mm), as shown in Figure 5. Several studies reported that ovigerous females were found with CW sizes between 4.0 and 5.2 mm, although some egg-bearing *D. japonicus* were observed to have CW as small as 4.0 mm. Fukui et al. reported that the maturity size of *D. tondensis* in the Tonda River corresponded to a CW over 4.0 mm [26]. Hiu et al. (2004) suggested that females with CW smaller than 5.2 mm may be functionally immature [14]. In our study, the smallest sizes of ovigerous females suggested that morphological sexual maturity occurred in small-sized individuals. Mature females were estimated to have CW larger than 3.9 mm. Also, no external sex differences could be observed up to 2.7 mm CW.

Reproduction is an important process in regulating marine benthic communities with complex life cycles and is determined by physical and biological processes that occur during the pelagic larval stage, settlement, and juvenile crab stage [33]. Also, the reproductive characteristics of these species are a result of the interaction between biological (food and food availability and competition) and environmental factors (temperature, salinity, photoperiod, rainfall, lunar cycles, and current) [34]. Temperature and salinity are known to be important factors that influence the distribution, survival, molting, and life cycle of crabs [10,35]. Rainfall is known to cause changes in the salinity of water and to promote an increase in the nutrients of planktonic larvae, which can reduce food competition in

larval stages. These conditions could determine the periodicity and extension of the reproductive period of a species, as well as its fecundity [36,37]. *D. japonicus* larval development was observed to consist of five zoeal stages and a megalopa. Very recently, we estimated that depending on the water temperature, salinity, and food availability encountered during larval development, *D. japonicus* larvae required about 23–27 days at 23–25 °C from hatching to megalopa [10].

The population of this species is found in the upstream, brackish waters of the Kita River estuary. The adult population of *D. japonicus*, including ovigerous females (96.0% of total ovigerous females), live mostly in the 4.4–6.8 km range, as optimal spawning sites. The areas of 4.8–6.4 km from the Kita River mouth are characterized at least seasonally by a higher relative abundance of early juvenile *D. japonicus* crabs as newly recruited individuals (Figure 8). The pelagic larvae of many estuarine species (e.g., brachyuran crabs) are conveyed away from adult habitats using selective tidal-stream transport. The larvae of many brachyurans develop for approximately of 1–2 months in the lower regions of bays, estuaries, and continental shelves and must be transported back to the adult habitats [38,39]. Their development is usually composed of four to five zoeal stages and a megalopa [20]. Furthermore, these unique characteristics may be illustrated in the differences in the population structure and biology between the adult and larval stages. As observed in other brachyuran crabs, the newly hatched larvae of *D. japonicus* aggregate in surface waters during flood tide. After a 3–4 week period of growth and development in the open sea or estuaries, the megalopa stage molts and sets out as a juvenile crab in the adult crab habitat.

Several authors have reported on the breeding season of this species. In a study on the Kita River estuary by Hiu et al., the reproduction of *D. japonicus* was reported to occur mostly from spring (March) to late summer (September) with the peaks of ovigerous females [12]. Fukui et al. reported that the reproduction of *D. tondensis* in the Tonda River occurred from April to November, with the peaks of breeding periods between June and July [26]. In the present study, ovigerous females in the Kita River estuary were found throughout the year, with breeding period peaks between July and September. The recruitment of *D. japonicus* was continuous throughout the year as a result of the high reproductive activity observed in the population. However, Figure 7 shows that the peaks of young recruits were found between September and November. Fukui et al. suggested that the peaks of young recruits occurred around late summer [26]. In this species with continuous breeding and recruitment, we expected to see relatively stable frequency distributions throughout the year. Reproductive activities and breeding periods are shorter at high latitudes and longer at low latitudes. Fukui et al. suggested that the occurrence of an extensive breeding period occurred from spring to early winter in the Tonda River (33°38′ N, 135°24′ E). In contrast, reproductive activities and breeding periods of females were found continuously throughout the year in the Kita River. According to Emmerson, most tropical species have a continuous reproductive period throughout the year or have longer reproductive seasons than higher-latitude species [30]. Crane suggested that continuous reproductive activity is common for fiddler crabs, which are typically adapted to living in warm regions [40]. Also, the water temperature has been established as a factor promoting growth and early ovarian development [41]. Mean water temperatures are slightly higher in the Kita River than in the Tonda River. Therefore, we assume that the water temperature could be the main factor for the latitudinal differences observed in the breeding periods of this species. Another factor could be the local feeding conditions. Although we do not have data for this factor, local food availability or quality has been demonstrated to influence early ovarian development in decapods [42].

Fecundity may vary with regard to the latitudinal range, habitat, and food availability [43]. Fecundity is determined not only by the female body size but also by the average EN and the brood frequency. Fukui et al. insisted that the fecundity of *D. tondensis* increased isometrically by the slope of the logarithmic equation between the EN and the CW [26]. Also, *D. tondensis* showed a high reproductive effort (four to five broods) per year. In marine and brackish water species, the egg size varies mostly between 250 and 450 µm, without showing a relationship with specific ecological traits [20]. *D. japonicus* has large eggs (430 µm in diameter). The mean fecundity was estimated to

be 1008.3 ± 183.1 (mean ± SD: 8.3 ± 1.6 mm) eggs per female and was directly related to the CW of ovigerous females. The relationship between the EN and the CW was described by the function EN = 110.36 × CW + 90.96. The determination coefficient ($R^2$ = 0.948) had a higher value than that reported by Fukui et al. ($R^2$ = 0.826) [26]. These results explain the low fecundity per brood found by those authors [20,26]. Also, the number of eggs in *D. japonicus* increased in accordance with CW, as found in other brachyurans. Several females with the same CW showed differences in the number of eggs carried. This fact may be related to food availability, environmental conditions, and the natural loss of eggs. Our results suggest that the fecundity of this species is influenced by the CW in females and by biological factors during the reproductive periods.

## 5. Conclusions

*D. japonicus* is one of near-threatened species which is known to reside in isolated locations and in upstream, brackish waters from Kanagawa Prefecture to Okinawa Prefecture in Japan. The present study described the distribution and reproductive biology as well as the recruitment pattern of juveniles and the life cycle of *D. japonicus* in the Kita River. Field sampling for this study was performed at bimonthly intervals for a long time of approximately nine years, from May 2001 to November 2008 and from November 2014 to January 2016. The population structure of *D. japonicus* during the sampling periods suggested a relationship with the environmental factor of habitat in the Kita River. Moreover, this information is important in designing adaptive management strategies for the conservation of this species. Finally, we can conclude that *D. japonicus* would be an indicator for environmental evaluation in the Kita River, Japan, and such periodic survey research provides a methodology to understand the habitat of endangered species in the environment.

**Author Contributions:** I.-K.O. planned the experiment and collected and analyzed the data. I.-K.O. prepared the manuscript, and S.-W.L. helped in drafting the manuscript and in the interpretation of the results. All authors contributed to revisions and completion of the manuscript. All authors have read and agreed to the published version of the manuscript.

**Funding:** This research received no external funding.

**Acknowledgments:** This project was supported by the River Ecology Group of Japan and from the Fundamental Science Research Program of the Japanese Ministry of Science and Education. We would like to thank several students at the graduate school of Urban and Environmental engineering, Kyushu University for field survey.

**Conflicts of Interest:** The authors declare no conflict of interest. We have already mentioned funding sources in acknowledgment section; no other relationships or activities that could appear to have influenced the submitted work.

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
