# Peer review of "Population Structure and Reproductive Biology of the Endangered Crab Deiratonotus japonicus (Brachyura, Camptandriidae) Surveyed for Nine Years in the Kita River, Japan"

_jmse, doi:10.3390/jmse8110921_

Round 1

Reviewer 1 Report

Review “Population structure and reproductive biology on endangered crab….”
October 2020

This study is a long-term study on the population biology and reproductive biology of the endemic crab.

The authors present much data in an organized fashion and the paper is very well-written. I am attaching a PDF with comments and also organizing comments below:

1) In the materials and methods:
a. Ovigerous condition and how they determined this needs to be clearly stated as ovigerous females with eggs is not analogous with o.condition.
b. The map is confusing. The other O. Prefecture is missing so the sampling area is not clearly defined.
c. Are the 38 stations in this 7 km stretch?
d. In the abstract, it says the sampling area is 5.2 km so need to be clearer where this in on the map. Because the distance from the river mouth is so crucial to this study, having a very clear location of sampling area is needed.
e. Need to state if males and non-ovigerous females were released, preserved, etc.? It is not mentioned so it seems as if individuals could be collected later on during sampling which would make numbers maybe inaccurate.
f. Line 78, a citation is needed for the statement as it is in the methods.

2) In the results:
a. Figure 3: needs to be divided into images A-F with labels and then the figure legend needs to cover each of these images explaining what each are.
b. Did not mention if sizes of individuals varies from the river mouth—competition for space at a preferred salinity?
c. What is meant by line 136 and “no clear sex”—give the size ranges for these animals very clearly.
d. In the Table 1.1: months are not consistent throughout the years (i.e. July and January—some of the months are higher with females and then next year lower—with a consistent trend)—need to address this in results and then discussion (Why this may be occurring)—I would have expected the same sex ratios in the same months over the years unless something is forcing it to change.
e. Make sure all data is given in the results as in the discussion (line 214) some data seems to presented in a new way or just new.
f. Topics to make sure are discussed in the discussion based on the results (they currently are not):
i. Link sex ratio and density with distance from the river and why the authors
think that this is occurring.
ii. Line 131—regarding moving downstream and changes—why and what is the
benefit to the crabs.
iii. Figure 8-reason why further down from mouth juveniles and female population changes

g. In the discussion:
i. Great line 245 paragraph—please also discuss this with females and ovigerous

ii. Paragraph 253 is a bit short—consider combining with the paragraph above or below.

Author Response

Dear Reviewer

Thank you for your comments and kind suggestion.

Please find the attached file including our responses to your comments.

Sincerely yours

Seung-Woo Lee  

Reviewer 2 Report

In this manuscript, the authors describe the population structure and the reproductive biology of an endangered and endemic crab species (Deiratonotus japonicus), based on 9 years of survey. They recorded the sex-ratio (global and over time), the density of individuals, the carapace width (related to the female ability to produce eggs), and they placed those informations in an ecological and a conservation context.

This is an interesting and valuable study to increase our understanding of an endangered and poorly studied clade. The manuscript is well written and organized. The reviewer has only several minor concerns, which are listed below following the order of the manuscript.

Line 47 : "  Several aspects[...]" : a large space is visible between the "." from the previous sentence and the word "Several". Probably a typo to fix.

Line 68-70 (Figure 1) : The map is theoretically refering to the 38 sampling stations of the study. However, these stations are not present in the map... Is it possible to add the stations on the map ? Or may be it will be necessary to reconsider the legend " Map showing sampling stations [...]". Because the stations are not visible on this map (the same as the Figure 1 from one of your latest paper "Effects of Temperature on the Survival and Larval Development of Deiratonotus Japonicus (Brachyura, Camptandriidae) as a Biological Indicator", may be a mistake ?)

Line 87/88 : The authors used a chi-squared to evalute the deviation from a balanced sex-ratio. I think it is better to use a binomial test instead of. the binomial test is an exact test of the statistical significance of deviations from a theoretically expected distribution of observations into two categories (typically in your case : a 1:1 sex-ratio). The result should be the same as those produced by the chi-squared tests, but it is more statistically correct to use binomial tests. It really easy to perform using R software for example (binom.test() function).

Line 89/90 : The authors used a Komolgorov-Smirnov test to test he null hypothesis that male and female samples had similar distributions (normal distribution ?). Classically, it is also needed to test the homoscedasticity of variance (Levene or Bartlett test) in addition to assess the distribution of the data, before using parametric statistical tests.

Line 129/130 : The authors explained that "D. japonicus has shown a tendency to slightly move downstream recently when compared with its distributions during the first sampling period". To me, it is not really obvious that this tendency is really occurring (considering the Figure 4). Do you have any way to test that tendency (in a statistical way) ? Or at least to precise the p-value obtained by the KS test (since it is a tendency, the p-value should be not that far from 0.05, no ?)

Line 145 (Table 1.1) : I don't understand why is it written "-2" in the number of ovigerous females recorded in May 01 (first line of the table)... What are these "negative ovigerous females" ?

Line 184 : The number "8"  from "p> 0.05, Figure 8)" is highlighted in yellow, probably another typo here.

Line 245-252 : the police size of these 2 paragraphs appears smaller than the other parts of the manuscript on my computer... just be sure that everything is fine here.

Author Response

(The authors gave the same response as above.)

Reviewer 3 Report

Dear Authors

Present work "Population structure and reproductive biology of the endangered crab Deiratonotus japonicus (Brachyura, Camptandriidae) surveyed for nine years in the Kita River, Japan" could have a potential interest due to the extensive sampling time period. However, I found grave mistakes in methodology and statistics. Inapropiate result presentation and contradictory results. I would be respectful with the authors, and in my opinion a great effort was made, but authors should contact with an expert in crabs biology.

Some commnets:

Lines 98-100. The second sentence indicates just the opposit of the first result. 

Kind Regards

Author Response

(The authors gave the same response as above.)

Round 2

Reviewer 3 Report

Journal: J. Mar. Sci. Eng.

Manuscript ID: jmse-969071

Title: "Population structure and reproductive biology of the endangered crab Deiratonotus japonicus (Brachyura, Camptandriidae) surveyed for nine years in the Kita River, Japan" 

Author(s): Il-Kweun Oh*, Seung-Woo Lee

Reply to Reviewer #3

Present work "Population structure and reproductive biology of the endangered crab Deiratonotus japonicus (Brachyura, Camptandriidae) surveyed for nine years in the Kita River, Japan" could have a potential interest due to the extensive sampling time period. However, I found grave mistakes in methodology and statistics. Inappropriate result presentation and contradictory results. I would be respectful with the authors, and in my opinion a great effort was made, but authors should contact with an expert in crabs biology.

[Authors]: We appreciate Reviewer's positive consideration and meaningful opinion for our article.

Comment 1. Lines 98-100. The second sentence indicates just the opposite of the first result. 

[Authors]: In response to the referee’s comment, we omitted the sentence on line 99 “The abundance of males illustrates their predominance over females (Figure 2).”

Thanks for consider my suggestion.

Second round

Dear Authors I apologies for the lack of feedback in my first review. As I said in my  first revision I found mistakes in the methodology that made myself to recommend the rejection of your work. All the comments are conducted in a constructive way. I’ll try to introduce my main concerns about some aspects of the present document.

My first concern is about the density of crabs presented in results (Line 131) and figure 4. The density term appears for the first time in the main body at results sections (excluding the abstract). I miss the explanation about how authors calculate the density of crabs in material and methods. To my knowledge density of crabs captured through a trap only could be calculated from the trap effective fishing of the area (EFA). This estimation require a previous experiments or a previous studies in a similar traps whit the same bait. I greatly appreciate an explanation about how this result has been obtained. At the same time,  I miss the bait used to capture the crabs.

My second concern is about the fishing effort and the number of traps usen in each station. I concluded from the material and methods that a single trap was depleted in each station. If it was the case, authors did made any replicate. I greatly appreciate a clear explanation about the effort of captures through the sampling period and  specially if it was the same during all the study (i.e missed traps, damaged traps, traps with predators).

Kind regards

Author Response

Dear Reviewer,

Please find the attached file including our response to the comments.

Thank you in advance

Sincerely yours

Round 3

Reviewer 3 Report

Journal: J. Mar. Sci. Eng.

Manuscript ID: jmse-969071

Title: "Population structure and reproductive biology of the endangered crab Deiratonotus japonicus (Brachyura, Camptandriidae) surveyed for nine years in the Kita River, Japan"

Author(s): Il-Kweun Oh*, Seung-Woo Lee 

Reply to Reviewer #3 (3nd Round)

Dear Authors,

There is my response in red

Comment 1. My first concern is about the density of crabs presented in results (Line 131) and figure 4. The density term appears for the first time in the main body at results sections (excluding the abstract). I miss the explanation about how authors calculate the density of crabs in material and methods. To my knowledge density of crabs captured through a trap only could be calculated from the trap effective fishing of the area (EFA). This estimation require a previous experiments or a previous studies in a similar traps whit the same bait. I greatly appreciate an explanation about how this result has been obtained. At the same time, I miss the bait used to capture the crabs.

[Authors]: We apologize for our unclear description. The commented issue associated with the estimation od the density of crabs captured using a trap may depend on how to capture them. However, as mentioned on lines 63 to 65 in the manuscript, the crabs were collected using a quadrat trap with a net (0.5 m ×0.5 m× 0.3 m). All the samplings conducted at those 38 stations were the same and no bait was used for sampling. The mean density of the crabs captured at each station was obtained by averaging the number of individuals per unit area where 2 to 5 collections were performed. For a clear understanding of readers, we added the highlighted sentence above in the revised manuscript. Also, the following photos show a normalized quadrat trap and sampling images in the field.

Reviewer 3

I really appreciate the pictures provided by the authors and I extrongly recommend the inclusion of the cuadrant tap picture  and the sampling demonstration both. To my knowledge, a trap is a passive sampling method usually with a bait. In my opinion the present sampling method is a core modified with a net. I sudgest to the authors the incluison (as a reference) of the other work  using the same sampling system with the same nomenclature. In my case, I was confused by the methodology. The methodology needs to be described with greater precision, in order to ensure reproducibility.

 Comment 1. My second concern is about the fishing effort and the number of traps usen in each station. I concluded from the material and methods that a single trap was depleted in each station. If it was the case, authors did made any replicate. I greatly appreciate a clear explanation about the effort of captures through the sampling period and specially if it was the same during all the study (i.e missed traps, damaged traps, traps with predators).

[Authors]: We appreciate Reviewer’s careful check and advice. Yes, as mentioned above, we used normalized traps of the same dimensions throughout all samplings to ensure the accuracy of sampling. This information has also been added to the revised manuscript.

The authors have answered my concerns

Kind regards

Reviewer 3

Author Response

Dear Reviewer,

Thank you so much for your helpful comment. Please confirm our response to the reviewer's comment.

Sincerely yours
